# Rare Causes of Arterial Hypertension and Thoracic Aortic Aneurysms—A Case-Based Review

**DOI:** 10.3390/diagnostics11030446

**Published:** 2021-03-05

**Authors:** Svetlana Encica, Adrian Molnar, Simona Manole, Teodora Filan, Simona Oprița, Eugen Bursașiu, Romana Vulturar, Laura Damian

**Affiliations:** 1Department of Pathology, “Niculae Stancioiu” Heart Institute Cluj-Napoca, 19-21, Calea Moților St, 400001 Cluj-Napoca, Romania; s_encica@yahoo.com; 2Department of Cardiovascular Surgery, “Niculae Stancioiu” Heart Institute Cluj Napoca, 19-21, Calea Moților St, 400001 Cluj-Napoca, Romania; adimolnar45@yahoo.com (A.M.); simona_oprita@yahoo.com (S.O.); 3Department of Cardiac and Thoracid Surgery, Iuliu Hatieganu University of Medicine and Pharmacy Cluj Napoca, 8, Victor Babeș St, 400012 Cluj-Napoca, Romania; 4Department of Radiology, “Niculae Stancioiu” Heart Institute 19-21, Calea Moților St, 400001 Cluj-Napoca, Romania; simona.manole@gmail.com (S.M.); ginobursasiu@yahoo.com (E.B.); 5Department of Radiology, Iuliu Hatieganu University of Medicine and Pharmacy Cluj Napoca, 8, Victor Babeș St, 400012 Cluj-Napoca, Romania; 6Department of Cardiology, Emergency County Hospital Alba, 23, Revoluției 1989, 510077 Alba-Iulia, Romania; margineanteo@gmail.com; 7Department of Molecular Sciences, Iuliu Hatieganu University of Medicine and Pharmacy Cluj-Napoca Romania, 6, Pasteur, 400349 Cluj-Napoca, Romania; 8Cognitive Neuroscience Laboratory, University Babes-Bolyai, 30, Fântânele St, 400294 Cluj Napoca, Romania; 9CMI Reumatologie Dr. Damian, 6-8 Petru Maior St, 400002 Cluj-Napoca, Romania; ldamian.reumatologie@gmail.com; 10Department of Rheumatology, Center for Rare Autoimmune and Autoinflammatory Diseases, Emergency Clinical County Hospital Cluj, 2-4 Clinicilor St, 400006 Cluj-Napoca, Romania

**Keywords:** thoracic aortic aneurysm, *PRKG1* mutation, PKG-1β isoform, hypertension, smooth muscle cells, scoliosis

## Abstract

Thoracic aortic aneurysms may result in dissection with fatal consequences if undetected. A young male patient with no relevant familial history, after having been investigated for hypertension, was diagnosed with an ascending aortic aneurysm involving the aortic root and the proximal tubular segment, associated with a septal atrial defect. The patient underwent a Bentall surgery protocol without complications. Clinical examination revealed dorso–lumbar scoliosis and no other signs of underlying connective tissue disease. Microscopic examination revealed strikingly severe medial degeneration of the aorta, with areas of deep disorganization of the medial musculo–elastic structural units and mucoid material deposition. Genetic testing found a variant of unknown significance the *PRKG1* gene encoding the protein kinase cGMP-dependent 1, which is important in blood pressure regulation. There may be genetic links between high blood pressure and thoracic aortic aneurysm determinants. Hypertension was found in FBN1 gene mutations encoding fibrillin and in *PRKG1* mutations. Possible mechanisms involving the renin–angiotensin system, the role of oxidative stress, osteopontin, epigenetic modifications and other genes are reviewed. Close follow-up and strict hypertension control are required to reduce the risk of dissection. Hypertension, scoliosis and other extra-aortic signs suggesting a connective tissue disease are possible clues for diagnosis.

## 1. Introduction

Thoracic aortic aneurysms (TAA) (defined as local aorta dilatations with a diameter increase of more than 50%) may result in dissection with fatal consequences if undetected [1]. TAA are a cause of early mortality, inasmuch as they are often nonsymptomatic [2]. The ascending and descending TAA differ due to differences in the embryonic origin of the cells involved in remodeling [3]. The thoracic and abdominal aortic aneurysms (AAA) also have different clinical characteristics, although they share a proteolytic degeneration of elastic tissue and vascular smooth muscle cells (SMC). However, unlike AAA, TAA are in 20% of cases inherited, transmitted in an autosomal dominant manner due to single-gene mutations [4]. Syndromic TAA, which can explain 5% of cases, are found in Marfan’s syndrome, Ehlers–Danlos and Loeys–Dietz syndrome, while nonsyndromic TAA refer to cases with familial history but without the typically associated features. Other associations of TAA are aortic bicuspidia and cerebral aneurysms. Acquired causes of TAA are inflammatory, due to aortitis in Takayasu arteritis, giant cell arteritis, ankylosing spondylitis, Crohn’s disease, sarcoidosis, IgG4 disease and others, or infections such as syphilis [5,6]. Additionally, fluoroquinolones have been recently associated with aortic aneurysm or dissection, probably by their affecting collagen cross-linking or metalloproteinase activities [5]. 

Most familial cases are monogenic, genetically heterogenous and autosomal dominant with decreased penetrance and variable expression [7]. Mutations interest the genes that encode proteins involved in vascular smooth muscle cell (SMC) contraction, elastin and microfibrils, the TGF-β pathway, the extracellular matrix (ECM) and other genes [2]. However, the list is increasing, and many unresolved nonsyndromic cases suggest the existence of undiscovered causal genetic variation [7]. 

Protein kinase cGMP-dependent 1 (PKG1) is a downstream effector of cGMP1, inducing SMC relaxation in response to nitric oxide (NO) signaling, which is important in blood pressure regulation [8]. 

From a clinical point of view, the incident diagnosis of a TAA should prompt a search for etiology and complications [9]. Recommendations have been developed for the cardiogenetic care of patients with TAA and their first-degree relatives [10]. 

We describe a case of TAA in which a young patient with no relevant familial history was investigated for early-onset hypertension, in whom a variant of unknown significance of the gene encoding PKG1 was found. 

## 2. Case Presentation

A 23-year male patient was referred to the Cardiovascular Surgery Department after having been investigated for an arterial hypertension, which had been detected since the age of 19, with values ranging from 140/80 mmHg to 160/70 mmHg. The renal ultrasonography had been normal, as were the urinary functional tests, catecholamines, aldosterone, thyroid hormones and renin. He had been smoking (seven cigarettes/day) for 4 years. His therapy included nebivolol (5 mg/day), perindopril plus indapamide (5 mg/1.25 mg/day) and ivabradine (5 mg twice/day) (prescribed for an episode of sinus tachycardia and transitory prolonged PR interval). His familial history was positive for arterial hypertension and dyslipidemia, found in both parents, gouty arthritis in his father and seronegative rheumatoid arthritis in his mother. No other aortic aneurysms were identified in the family.

Clinical examination revealed a height of 173 cm and a weight of 75 kg (a BMI of 25.4—overweight), dorso–lumbar scoliosis, flat feet, bilateral genu valgum and subluxating patellae, with no other signs of hypermobility, no cutaneous striae or other signs of underlying connective tissue disease. No signs of arthritis or enthesitis were present, either. A diastolic bruit in the aortic focus was heard. All preoperative tests were within limits, including cholesterol, hepatitis B and C antigens and HIV, C-reactive protein (CRP) (<6 mg/dl), erythrocyte sedimentation rate (ESR) (2 mm/h) and the neutrophil to lymphocyte ratio (NLR) (2.3). 

Echocardiography revealed a tricuspid aortic valve with annular dilation, a moderate/severe aortic insufficiency (vena contracta 6 mm and pressure half-time PHT 360 msec), a grade II mitral insufficiency, grade II pulmonary insufficiency, an interatrial septum aneurysm and an aneurysm of ascending aorta (Figure 1a,b). 

The chest X-ray showed an enlarged mediastinum due to ascending aorta dilatation and dorsal scoliosis (Figure 2).

The angio CT aortic scan depicted an aneurysm of the ascending aorta involving the aortic root and the proximal tubular segment, with loss of the sinotubular junction, a maximum diameter of 64 mm (near the pulmonary artery trunk) and no signs of rupture, associated with an interatrial septum aneurysm and septal atrial defect type ostium secundum (Figure 3a–c). 

The aortic valve was tricuspid. A common ostium for the brachiocephalic artery trunk and the common carotid artery was noted. All other arteries, including the renal artery, were normal; the kidneys and adrenal glands were normal, as well. He underwent a Bentall surgery protocol (the replacement of ascending aorta with a tubular dacron prosthesis nr 28, aortic valve prosthesis with a Carbomedics mechanic valve and reimplantation of coronary ostia). The postoperative evolution was with no complications. 

The aortic biopsy showed a thoracic aortic aneurysm with a diameter of 5 cm and a length of 6 cm; the rest of tubular aortic wall had a diameter of 2 cm (Figure 4).

Microscopic examination employed usual and special stains, hematoxylin–eosin (HE), orcein for the elastic laminae, trichrome Masson for the SMC and collagen fibers and Alcian blue/PAS for the ECM. A strikingly severe medial degeneration of the aorta was seen, with areas of deep disorganization in the medial musculo–elastic structural units and mucoid material. In HE staining, band-like areas with a loss of aortic SMC nuclei were noted (Figure 5a). Structurally, the medial wall showed significant changes due to the loss of elastic laminae, the focal collapse of the dense lamina and the disorganizing of the elastic laminae (Figure 5b,c). The aortic media contained intralamellar and translamellar puddle-shaped accumulations of mucoid material, severely altering the lamellar unit arrangement (Figure 5d). The SMC were also reduced in amount and disordered in the aortic media, with bundles of SMC with conspicuous disarray (Figure 5c). Vasa vasorum were obvious and slightly increased in amount in the external part of the medial layer (Figure 5a).

Perivascular T cells were found near the adventitial border of the media; macrophages in some mucoid material ponds and scattered foamy cells were noted as well. 

The genetic examination (sequence analysis and deletion/duplication testing of 27 aortopathy genes, conducted using Invitae Diagnostics (San Francisco, CA, USA) revealed a heterozygous variant of unknown significance (VUS) of the *PRKG1* gene (c.790A>G, p.Ile264Val) in exon 6, not reported in the literature in *PRKG1*-related diseases, present in population databases (rs56082459, ExAc 0.02%), with an allele count higher than expected for a pathogenic variant (PMID: 28166811). The sequence corresponds to isoform beta of the enzyme. Algorithms developed to predict the effect of missense changes on protein structure and function (SIFT, PolyPhen-2, Align-GVGD) suggest that the variant is likely to be tolerated, but this has not been confirmed by published functional studies, therefore necessitating the classification of the variant as a VUS.

The therapy after surgery consisted of 75 mg of aspirin, 4 mg of acenocumarol (with INR adjustment—target 2–3), 5 mg/day of perindopril and 20 mg/day of famotidine. The patient was advised to quit smoking and to lose weight, and was scheduled for regular clinical and imaging follow-up. 

## 3. Discussions

The proper aortic function is the result of a tightly regulated and highly preserved microstructural architecture of the aortic wall [11]. The aortic medial layer is composed by elastic laminae (60%), SMC (30%) and glycosaminoglycans from the extracellular matrix (1–5%). The medial lamellar units consist of two paired elastic laminae, encompassing smooth muscle cells, adhesion molecules, collagen fibers and glycosaminoglycans. There are about 50–70 alternative, concentric layers of elastic laminae and SMC [12,13]. Aortic SMC serve as mechanosensors through the elastin–contractile units, turning mechanic stimuli into biochemical signals to activate contraction [4]. 

The elastin–contractile unit links elastin to the SMC and consists of elastin fibers, surrounded by microfibrils (that bind to the integrin receptors in focal adhesions on the SMC surface) and SMC contractile filaments linked to the focal adhesions on the inner side of the surface [14]. The elastic fibers, stored primarily in the aortic wall during the peripartum period, normally have a long half-life (over 50 years). Degradation of the elastic fibers causes irreversible changes, not only in the structure of the aortic wall but also in its functions [12,15].

Aortic SMC are directly involved in establishing, maintaining and restoring aortic structural integrity. The contractile dysfunction of SMC is centrally involved in the occurrence of human TAA. Normally, the SMC elongate their shape and change the expression of specific contractile proteins, produce small amounts of ECM proteins and have minimal proliferation and migration. Glycosaminoglycans and proteoglycans in the aortic ECM, collagen III fibers in the media and collagen I in the adventitia help maintain the structure and function of the aortic wall [12,15].

Damage to the integrity of contractile elastic units, either by conformational changes in genetic mutations or by mechanical pressure aggression (as in hypertension), affects the sensing function and the contraction of SMCs, resulting in thoracic aneurysms [16]. Aortic aneurysmal transformation involves TGF-β [15,17]. Notably, SMC density is preserved despite the thinning of the wall, degeneration and the loss of SMCs, suggesting SMC autophagy and hyperplasia [18]. 

According to the working consensus adopted by the Society of Cardiovascular Pathology and the Association for European Cardiovascular Pathology, degeneration of the aortic media depicts the summary of several histological characters describing the changes in the musculo–elastic unit (both at the cellular and extracellular level), seen in usual and special stains (for the elastic laminae, SMC, collagen fibers and the matrix material) [13,14]. The medial laminar collapse and loss of SMC nuclei in aortic media are the result of SMC apoptosis. The individual components of medial degeneration are as follows: intralamellar and translamellar accumulation of mucoid material, fragmentation or loss of elastic laminae, thinning of elastic fibers, disorganization of the elastic fibers, loss of SMC nuclei, medial laminar collapse, disorganization of the SMC and medial fibrosis. Medial aortic degeneration can be graded as mild, moderate or severe [13].

In our case, the severe medial degeneration evoked the damage of all the medial structural components, with SMC nuclei loss, areas of SMC hyperplasia with a disorganized, unusual, band-like disposition and dense laminar collapse and increased small arteries from the vasa vasorum. There were large areas of loss and fragmentation of the elastic laminae and significant multifocal extralaminar accumulation of mucoid material. This mucoid material compromises the aortic mechanosensing capability by mechanically separating the elastic laminae [17]. In genetic syndromes associated with TAA, the mucoid material is likely synthetized via increased TGF-β activity by SMCs that lose their contractile phenotype and acquire synthetic functions [17,19]. Genetic TAA appear to share the histological changes, which nevertheless differ in quantity [17,20]. By contrast, the medial degeneration in people over 50 years is due rather to aging and hypertension [21]. 

Aortas in *PRKG1* mutations have increased proteoglycan accumulation, decreased SMCs, elastic fiber loss and fragmentation and multiplication of small arteries in the medial layer from the vasa vasorum, like in our case [22]. Nevertheless, similar invasion of the vasa vasorum was noted in *MYH11* and *MYLK* mutations as well [22]. 

The presence of lymphoid elements near the small vessels is likely a consequence of the aortic media repair. Macrophage accumulation, through the production of cytokines and proteases, may initiate the weakening and possible delamination of the aortic media, favoring aortic dissection [19]. Of note, the SMC that modulate their contractile phenotype under injury may also acquire macrophage-like characters [17].

Inflammation in TAA is still controversial. Nevertheless, immunohistochemistry revealed CD3+ T cells and CD68+ macrophages that were more frequent in aneurysms than in normal aorta, as in our case [23]. Inflammation was found in Marfan’s syndrome, with the expression of inflammatory cytokines, the infiltration of inflammatory cells in the aortic wall, increased proteolysis and vascular SMC apoptotic markers [24]. Macrophages and T lymphocytes are found in sporadic TAA as well, and may contribute to SMC and ECM degradation, with possible therapeutic consequences [23]. A proinflammatory milieu may increase the risk of aneurysm progression [23]. Our patient had a familial history of inflammatory disease, but no current signs, symptoms or ancillary evidence of arthritis. 

The preoperative NLR has recently been described as a predictor of cardiovascular risk and adverse outcome [25]. NLR is associated with symptomatic and ruptured TAA and may alert clinicians for TAA complications [26]. 

### 3.1. Genetic Testing in TAA: The Role of PRKG1

Genetic testing in TAA may reveal mutations in the structural components of elastin and microfibrils (*FBN1, MFAP5, ELN* and *FBLN4*), in the smooth muscle cell contractile unit genes (*ACTA2, MYH11, FLNA, FOXE3, MYLK* and *PRKG1*), FLNA (linking integrin receptors to the contractile filaments), in the TGF-β pathway genes (*TGFBR1, TGFBR2, TGFB3, SMAD2, SMAD4, SMAD3, SMAD6, SKI* and *SLC2A10*), in genes in the extracellular matrix (*COL1A2, COL5A1, COL5A2, EFEMP2, ELN, EMILIN1, COL3A1, FBN1, FBN2, BGN, LOX* and *MFAP5*), and in other genes (*MAT2A, NOTCH1*) [2,27]. Protocols regarding gene- and aneurysm-specific interventions have been elaborated [2,27]. The majority of dissection cases have no pathogenic variant identified, either due to a pathogenic variant currently considered a variant of unknown significance (VUS), to a pathogenic variant in a yet not identified gene, to a polygenic aortopathy and/or to environmental factors [28]. 

Gain-of-function mutations of *PRKG1* (on chromosome 10) encoding PKG1 (which controls vascular SCM relaxation) have been described in association with TAA [29]. The protein PKG1 (cGMP-dependent protein kinase 1) is a serine/threonine protein kinase and acts as key mediator of the nitric oxide (NO)/cGMP signaling pathways; the abnormal enzyme PKG1 is constitutively activated even in the absence of cGMP binding, thus leading to the decreased contraction of vascular SMC, to TAA and to dissection at a young age (with a mean age of 31 years, ranging from 17–51 years) [22]. The pathogenic *PRKG1* mutations increase PKG1 activity, which inhibits myosin light chain phosphatase, functioning as a negative regulator of actin–myosin interaction [30]. 

The mechanisms resulting in TAA in *PRKG1* mutations have been recently described [31]. A hyperactive PKG1 leads to JNK (c-Jun N-terminal kinase) activation and upregulation of NOX4 (NADPH oxidase 4), leading to oxidative stress, consecutively initiating SMC apoptosis and metalloproteinase MMP-2 activation, which in turn result in the degradation of elastic fibers. Increased SMC apoptosis and impaired SMC proliferation cause SMC depletion. The secondary increase of TGF-β and NOX4-induced dysregulation of hypoxia-inducible factor-1 (HIF-1α)/vascular endothelial factor (VEGF) signaling further contributes to aneurysm progression [31]. 

The *PRKG1* pathogenic variants described to date are c.530>A (pArg177Gln) [7,29,32], and c.1108G>A (p.Gly370Ser)—the latter are associated with abdominal aneurysm, arterial tortuosity and keratoconus [33]. The c.530>A (pArg177Gln) mutation accounts for 1% of the nonsyndromic heritable thoracic aortic disease cases [32]. A recent multicentric study of the *PRKG1* c.530>A (pArg177Gln) mutation revealed more than 40% dissections of descending thoracic aorta in the 29 individuals carrying the mutation, with a mortality of 33% in the series at the median age of 24 years (range 19–43 years) [32]. The *PRKG1* mutation identified in our patient is a VUS with a substitution in the 264 position, involving the cGMP weak-binding domain (ref. UNIPROT, CLINVAR) (Figure 6).

The VUS in aortopathy genes have a controversial status, as they were not significantly associated with aortic pathology in a recent whole-exome sequencing study (found in 27.2% of cases versus 19.8% of controls) [28]. However, the risk of hereditary TAA associated with PRKG1 mutations is strong, based on the ClinGen Aortopathy Working Group classification. Moreover, VUS in aortopathy genes can actually be low-penetrant “risk variants” that may trigger disease in the context of other environmental, genetic or epigenetic risk factors [34,35]. In this setting, a genetic predisposition may be enhanced by other risk factors altering the contractile unit, such as hypertension, inflammation or other genetic mutations. 

Heritable thoracic aortic disease affecting the elastin–contractile unit, including PRKG1 mutations, may have extra-aortic features, such as hypermobility or cutis laxa, Marfan syndrome-like or Ehlers–Danlos-like features, without fulfilling the diagnostic criteria for these syndromes [14]. Scoliosis is not an infrequent finding, and it was present in our patient as well (Table 1). Of interest, *PRKG1* is a novel negative regulator of osteoblast differentiation in human skeletal stem cells, and *PRKG1* was found to have increased expression in idiopathic scoliosis, where it may become a therapeutic target [36,37]. 

### 3.2. Hypertension in TAA

Hypertension is an additional important factor for aneurysm or dissection in an aortic wall already fragile due to inherited or acquired defects [38]. Arterial hypertension may further alter the elastin–contractile unit proper function, leading to SMC apoptosis, to increased production of matrix metalloproteinases, proteoglycans and TAA [4]. 

Hypertension results from the interaction of multiple genetic and environmental risk factors, most of them involved in the handling of salt and potassium [39]. Both mechanical and humoral factors are stimuli for subsequent SMC hypertrophy in this setting [40]. 

There may be a link between genetic determinants of high blood pressure and TAA. In a large cohort followed up over 18 years on average, the genetic risk score for high blood pressure (based upon 29 single nucleotide polymorphisms), was significantly associated with TAA (with a hazard ratio of 1.64), but not with aortic artery aneurysm (AAA) or AA [41]. 

Hypertension was found in mutations of *FBN1* gene encoding fibrillin, which may evolve with marfanoid habitus [29,41,42]. In the Yale cohort, hypertension was found in 81.3% of cases with dissection and in 71% of cases of nondissecting TAA, versus 38% controls (the rs2118181 and rs10519177 were most associated with TAA and dissection) [43]. Additionally, in two large population studies, the G–A substitution in *FBN1* (rs11856553) was associated with moderate to severe prevalent hypertension [44]. On the contrary, in a large Chinese cohort, hypertension was more prevalent in patients without the *FNB1* mutation (30.4% of patients with *FBN1* mutations had hypertension versus 62.6% in the patients without the mutation), wherein the Ala27Thr substitution was more frequent [29]. In *PRKG1* mutations, hypertension was found in over 70% of cases in a series [32].

Fibrillin-1 is an extracellular matrix protein with load-bearing and anchoring functions in the arterial wall, directing the orientation of elastin fibers associated with aortic and arterial stiffness; *FBN1* genotypes may thus contribute to hypertension [45]. Recent data also show that fibrillin-1 induces endothelial cell apoptosis, inhibits their proliferation and contributes to vascular rarefaction and renal fibrosis in chronic kidney disease [46]. 

Another explanation could reside in the involvement of the renin–angiotensin system (RAS) in most inherited TAA [4]. All mutations decrease SMC contraction (*FBN1, ACTA2, MYH11, MYLK* and *PRKG1*) and activate angiotensin and TGF-β signaling as part of the cellular repair pathways [4]. RAS has multiple functions, including vasoconstriction, fluid volume regulation and others. The RAS plays a key role in hypertension, as binding of angiotensin II to its receptor AT1R results in vasoconstriction and the activation of cellular growth pathways mediated by tyrosine kinases [47].

PKG1, aside from vascular SMC contraction, is also involved in NO signaling, increasing Ca++ concentration in cells and, therefore, in blood pressure control [48]. An intriguing aspect regarding hypertension in *PRKG1* mutations is the fact that the gain-of-function of PKG1 should lead to SMC relaxation and hypotension [22]; however, many patients including ours have hypertension instead, suggesting the involvement of other mechanisms. 

In the vascular system, PKG1 mediates vascular tone, vascular SMC proliferation and differentiation, and endothelial cell migration and proliferation [49]. PKG1 regulates blood pressure by suppression of selective G-protein coupled receptor (RGS), phosphorylation of sarcomere proteins such as titin, the control of TGF-β downstream events, the regulation of transient receptor potential canonical channels and mechanosensing [8] (Table 2).

In hypertension, cyclic stretch from luminal pressure changes and ECM signaling regulate SMC myosin activity, and cationic ion channels are important in constriction control. Vascular SMC cellular changes are modulated by signaling pathways involved in vasoconstriction, cellular migration, SMC hypertrophy, apoptosis and inflammation [47]. PKG1 negatively regulates SMC cell cycle and migration via the disassembly of focal adhesions [50].

Osteopontin (OPN) (a proinflammatory, integrin-binding protein) characterizes the proliferating vascular SMC phenotype in the aneurysmal aortic wall and is also a prerequisite for vascular remodeling in hypertension [51]. OPN is regulated by cyclic mechanical strain and mediates neointimal formation and medial thickening [40]. Normally, increased PKG1 downregulates OPN to limit SMC migration (UniProt). Nevertheless, oxidative stress in a hypertension setting may increase OPN [40].

PKG1 is oxidized independent of cGMP, regulating blood pressure by micro-vascular dilatation [8]. Oxidation occurs at the cysteine 42 residues between the two homodimers in the N terminus, just above the leucine zipper that mediates PKG1 dimerization and substrate interaction [8]. Notwithstanding, PKG1 can also promote the formation of reactive oxygen species and oxidative stress, which is involved in hypertension [52].

An impaired PKG1 stimulates the vascular SMC phenotype switch from contractile to synthetic [50]. While SMC change their role, epigenetic modification, including chromatin remodeling, affect disease development and progression [53]. 

Inflammation may contribute to both genetically-induced TAA and inflammation. For instance, serum IL-6 levels were associated with aortic size in the GenTAC registry (Fujita). IL6 is also involved in Ang-II induced micro-vascular dysfunction and hypertension [54]. Moreover, TNF-α downregulates PKG1 through miR-155-p, inducing SMC phenotype and function alteration, which is important in inducing hypertension in inflammation [50,55]. Other epigenetic regulators such as miRNA 145 promote ascending aorta aneurysms via TGF-β1 and increase OPN and collagen expression [1]. 

Last, but not least, other genes may contribute to hypertension in PRKG1 mutations. For instance, a KEGG (Kyoto Encyclopedia of Genes and Genomes) pathway analysis of differentially regulated genes in Marfan syndrome revealed other genes involved in the cGMP–PKG signaling pathways, including *Irs2, Irs3,* (which encode insulin receptor substrates) and *Adcy7* (encoding adenylate cyclase 7) involved in the pathogenesis of hypertension [56]. Additionally, in hypertensive patients with TAA, the variant allele of THBS2 encoding the thrombospondin-2 (which may lead to an increase in metalloproteinase MMP-2) is a risk factor for TAA [57]. MMP-2, elevated in TAA, also mediates angiotensin-II induced hypertension [58,59].

**Table 2 diagnostics-11-00446-t002:** Possible mechanisms associated with hypertension in thoracic aortic aneurysm.

Mechanism	Effector	Molecular Interactions	References
Vasoconstriction	Renin–angiotensin system	Ang-II signaling activated by mutations decreasing VSMC contraction;	[4]
	PKG1	Key mediator of the NO/cGMP signaling pathways in VSMC contraction; Suppression of RGS, phosphorylation of sarcomere proteins; Control of TGF-β downstream events, regulation of transient receptor potential canonical channels and mechanosensing; Dysregulation of hypoxia-inducible factor -1 (HIF-1α)/ VEGF signaling;	[48] [50] [31]
Endothelial cell migration and apoptosis	PKG1	Regulation of endothelial cell migration;	[47]
	Fibrillin-1	Induction of EC apoptosis; inhibition of EC proliferation; Involvement in vascular rarefaction in chronic renal injury;	[46]
Vascular smooth muscle cell migration, apoptosis and phenotype switch	PKG1	Reduction of VSMC migration by focal adhesion disassembly; Regulation of VSMC apoptosis, proliferation, differentiation and phenotype switch from contractile to synthetic;	[8,31,49,50]
	Osteopontin	Induction of medial thickening and neo-intimal formation;	[40]
Extracellular matrix remodeling	Defective VSMC contraction	*FBN1, ACTA2, MYH11, MYLK* and *PRKG1* activate TGF-β signaling	[4]
	Fibrillin-1	Important in elastin deposition, anchoring and load bearing, associated with aortic and arterial stiffness	[45] [46]
	Metalloproteinases	MMP2 activation by hyperactive PKG1; Increased MMP2 involved in TAA and hypertension;	[31,57,58,59]
	Epigenetic regulators	miRNA 145 activates TGF-β and increases OPN and collagen expression;	[1]
Oxidative stress and inflammation	PRKG1	Formation of reactive oxygen species and oxidative stress by activating JNK and NOX4;	[31,52]
	Osteopontin	Upregulation by oxidative stress;	[40]
	Epigenetic regulators	miR-155-p used by TNF-α to induce VSMC phenotype and function alteration;	[50,55]
	Pro-inflammatory cytokines	IL-6 contributes to angiotensin II—induced microvascular dysfunction; IL-6 associated with aortic dimensions in TAA	[54] [60]

Legend: Ang-II—angiotensin-II; EC—endothelial cell; cGMP—cyclic guanosine monophosphate; HIF- hypoxia inducible factor 1 α, IL-6—interleukin 6; JNK—c-Jun N-terminal kinase; NO—nitric oxide; NOX4—NADPH oxidase 4; MMP-2—metalloproteinase 2; OPN—osteopontin; PRKG1—Proteinkinase cGMP-dependent 1; RAS—renin–angiotensin system; RGS—selective G-protein coupled receptor; TAA—thoracic aortic aneurysm; TGF-β—transforming growth factor β; TNF-α—tumor necrosis factor-α, VEGF- vascular endothelial growth factor, VSMC—vascular smooth muscle cells.

### 3.3. Risk of Acute Thoracic Aortic Syndrome in TAA According to Etiology

Acute aortic syndromes include aortic dissection, intramural hematoma and penetrating atherosclerotic ulcers [61]. The differences in aneurysms and their complications’ occurrence are likely due to the heterogeneity of aortic SMC origin [62]. Isolated ascending TAA tend to occur in younger patients with a genetically-triggered disease, while descending TA aneurysms are more frequently associated with hypertension, atherosclerosis and increased age [30,63]. In the absence of a mutation, arterial hypertension is the major etiological factor for TAA, and the pathway is common. In the follow-up, the descending TAA or mixed ascending and descending TAA have an increased risk of acute dissection or rupture and a higher mortality than isolated ascending TAA [63,64,65,66].

In genetic TAA, Marfan’s syndrome patients develop dissections, more likely at the aortic root and ascending aorta, and patients with truncated *FBN-I* mutations have higher risks of dissection at earlier ages [2,67]. Mutations in *ACTA2, MYLK, MYH11* and *PRKG1* are associated with dissection at aortic diameters under 5 cm dissection [2]. *PRKG1* mutations have a high prevalence of dissections (63%) involving not only the ascending, but also the descending aorta; the dissections were reported after 17 years, with an average age of 37 years, and the aortic diameter indicating prophylactic surgery is 4.5–5 cm [22,32].

Dissections in inflammatory noninfectious TAA were more often found in giant cell arteritis and Takayasu arteritis (which may involve any aortic segment, with a predilection for ascending aorta) [68,69]. In Takayasu arteritis, aortic dissection in TAA is associated with hypertension and patients may have atypical signs, with no history of chest or back pain, but fatigue and breathlessness instead [70]. Although IgG4 disease is generally more often associated with abdominal involvement, it was also reported in isolated TAA [71]. 

Pregnancy is associated with an increased risk of TAA dissection, mainly of the ascending aorta (60%), often associated with Marfan’s syndrome and hypertension [66,72].

Other types of thoracic aortic syndromes (such as intramural hematoma and penetrating aortic ulcers of an atherosclerotic plaque) occur primarily in older age and more often involve the descending aorta, more rarely having aortic valve insufficiency and aortic aneurysms [61]. 

### 3.4. Therapy of PRKG1-Associated TAA

The therapy of *PRKG1*-associated TAA is surgical, according to the guidelines and further monitoring for other aortic dilatations and aneurysms, including of the descending thoracic and abdominal aorta [32]. 

The medical treatment is based upon control of other factors which could disrupt the elastin contractile unit, mainly hypertension and other causes of systemic inflammation. Treatment of hypertension is paramount in all patients with TAA [65]. Therapies in TAA consist mainly of beta-blockers decreasing the blood pressure and the aortic wall’s rate of stretching. However, their long-term use may be limited in young patients by side effects such erectile dysfunction, which is rare with nebivolol [73]. Calcium channel blockers, ACE-inhibitors and angiotensin receptor blockers (losartan) may antagonize the TGF-β involved in aortic remodeling in TAA [74]. Of interest, N-acetylcysteine and an antioxidant vitamin B12 analog (cobinamide) were both recently shown to prevent age-related aortic dilatation in animal models [31]. 

Patients should be instructed to avoid the use of PKG1-activating drugs such as phosphodiesterase-5 inhibitors used for pulmonary hypertension or erectile dysfunction. Powerful stimulants such as cocaine may also increase the risk of aortic dissection [31]. Other modifiable risk factors for atherosclerosis (smoking, obesity and overweight, dyslipidemia etc.), should also be addressed. Patients are recommended to keep active while avoiding sports with an increased risk of aortic dissection such as weight lifting, isometric exercises etc. Attending clinicians and patients should be aware of early signs of aneurysm complications, including unusual presentations [65,75]. The screening of first-degree relatives and other family members with recording aortic calipers and genetic testing (wherever possible) is advisable

## 4. Conclusions

Detection of a TAA should prompt a search for etiology and testing of other family members. Hypertension and extra-aortic signs suggesting a connective tissue disease may be possible clues for the diagnosis. 

Hypertension in a young patient with a TAA may point to a search of *FBN-1, PRKG1* or other aorthopathy genes mutations. The *PRKG1* variant in the patient described is of unknown significance and was not confirmed as pathogenic. Nevertheless, VUS in aortopathies genes can be low-penetrant “risk variants” conferring a predisposition which may be enhanced by other environmental, genetic or epigenetic risk factors. Hypertension and inflammation are important risk factors that should be searched for and addressed in order to minimize risk [34,35].

A close follow-up with serial aortic measurements, assessment of aortic root dilatation and a strict hypertension control are required to reduce the risk of dissection. 

## Figures and Tables

**Figure 1 diagnostics-11-00446-f001:**
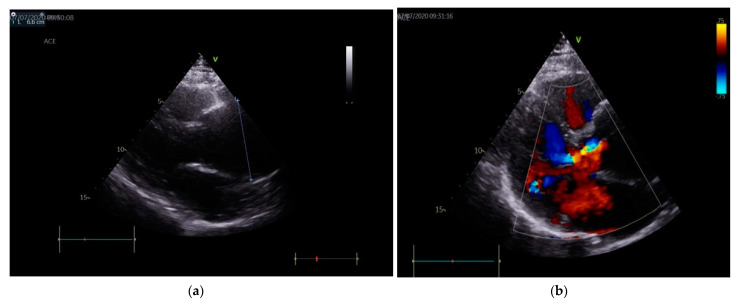
(**a**) Echocardiography (parasternal long axis view): aortic aneurism with diameter of 6 cm, indicated by the oblique line. (**b**) Color Doppler echocardiography: moderate/severe aortic regurgitation

**Figure 2 diagnostics-11-00446-f002:**
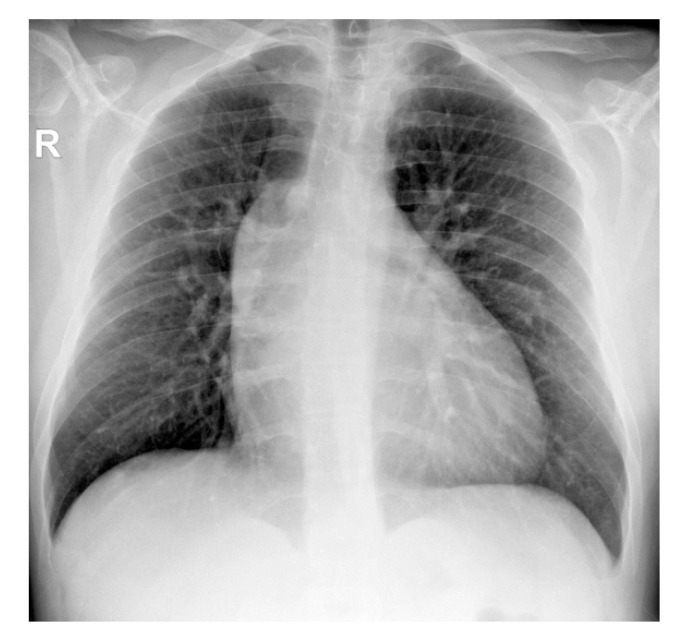
Thoracic radiography (AP projection)—widening of the middle mediastinum on the right (R); dextroconvex dorsal scoliosis.

**Figure 3 diagnostics-11-00446-f003:**
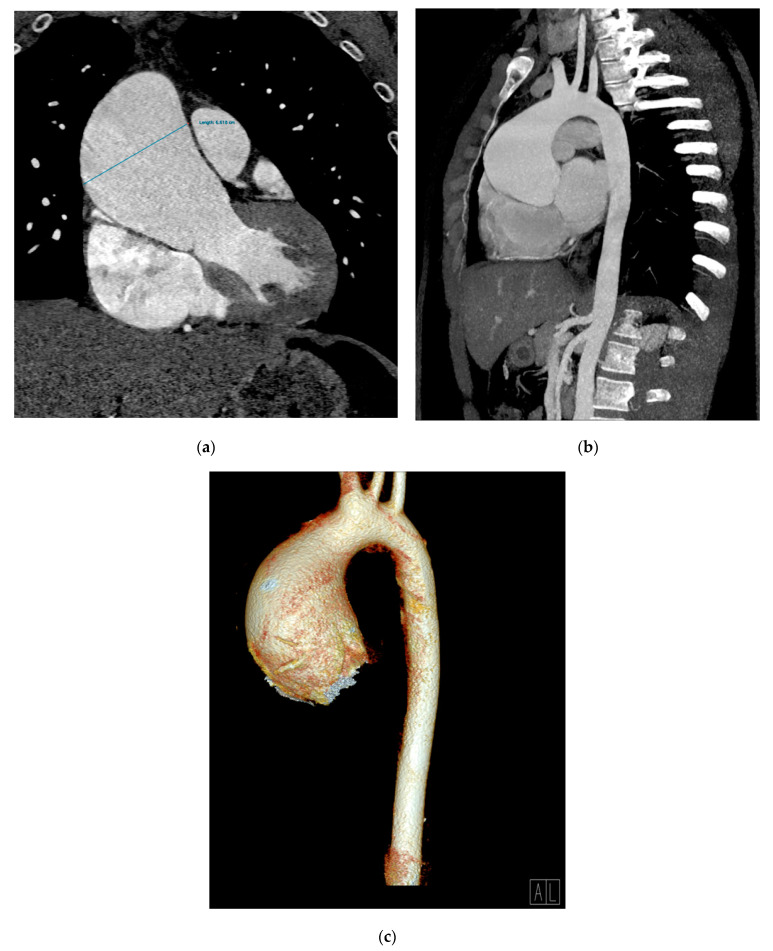
An angio CT aortic scan. Aneurysmal dilatation of ascending aorta (66 mm diameter); coronal multiplanar reconstruction (MPR) (**a**), oblique–sagittal maximum-intensity projection (MIP) reconstruction (**b**), and a 3D volume rendering technique (VRT) reconstruction (**c**).

**Figure 4 diagnostics-11-00446-f004:**
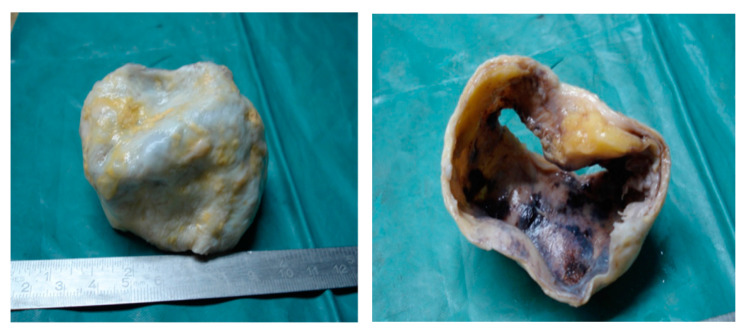
The resection piece—an ascending aortic aneurysm with a diameter of 5 cm.

**Figure 5 diagnostics-11-00446-f005:**
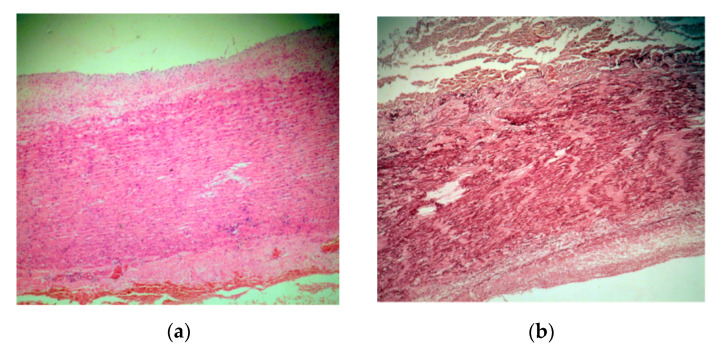
(**a**). A hematoxylin–eosin (HE) stain ×5 aorta with major medial changes, corresponding to severe medial aortic degeneration. There are the loss of smooth muscle cells nuclei, translamellar collection of mucoid material, and a small focal perivascular accumulation of lymphocytes towards adventitia. (**b**). Orcein stain ×10 elastic fiber loss, thinning, disorganization and laminar collapse. (**c**). Trichrome masson stain ×20 disorganized bundles of smooth muscle cells. (**d**). Alcian blue/PAS stain ×20 translamellar mucoid material accumulation.

**Figure 6 diagnostics-11-00446-f006:**
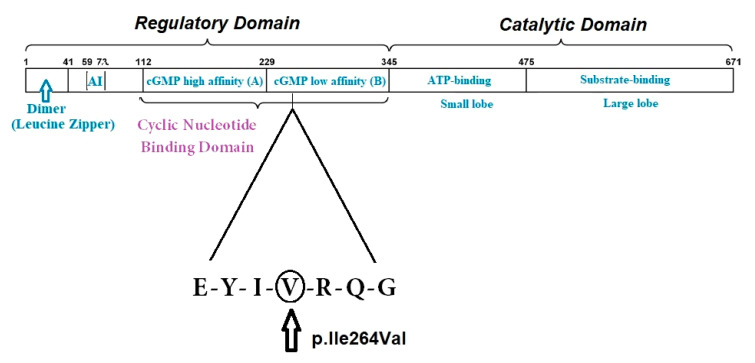
A graphic illustration of the functional domain structure of PKG-1β and the region where the missense variant p.Ile264Val is located, as indicated by the triangle. At the N-terminus, the dimerization domain (with the Leucine Zipper region) is depicted. AI = the auto-inhibition domain; cGMP high (called A) = the high-affinity binding domain for cGMP; cGMP low (called B) = the low-affinity binding domain for cGMP.

**Table 1 diagnostics-11-00446-t001:** Clinical features described in *PRKG1* mutations or variants.

Feature	Mutation or Missense Variant	Pathogenicity	References
Thoracic aneurysm	c.530G>A (p.R177Q) c.477C>T (p.Thr159Thr), c.993T>C (p.Val331Val), c.790A>G, p.Ile264Val	P VUS	[7,22,27], CC
Abdominal dissection	c.1108G>A (p.Gly370Ser)	P	[33]
Arterial tortuosity	c.530G>A (p.R177Q), c.1108G>A (p.Gly370Ser)	P	[22,33]
Hypertension	c.530G>A (p.R177Q), c.1108G>A (p.Gly370Ser) c.790A>G, p.Ile264Val	P VUS	[22,33] CC
Scoliosis	c.530G>A (p.R177Q) c.790A>G, p.Ile264Val	P VUS	[7] CC
Wrist and thumb sign	c.530G>A (p.R177Q)	P	[7]
Arachnodactily	c.993T>C (p.Val331Val)	VUS	[27]
Pectus carinatus/excavatum	c.530G>A (p.R177Q) c.993T>C (p.Val331Val)	P VUS	[7] [27]
Miopy	c.530G>A (p.R177Q) c.477C>T (p.Thr159Thr)	P VUS	[7] [27]
Keratoconus	c.1108G>A (p.Gly370Ser)	P	[33]
Small deep set eyes	c.993T>C (p.Val331Val), c.790A>G, p.Ile264Val	VUS	[27], CC
Skin striae	c.530G>A (p.R177Q) c.477C>T (p.Thr159Thr), c.993T>C (p.Val331Val)	P VUS	[7] [27]
Keloid scars	c.477C>T (p.Thr159Thr)	VUS	[27]
Premature aging	c.993T>C (p.Val331Val)	VUS	[27]
Subluxating patellae	c.790A>G, p.Ile264Val	VUS	CC
Joint pain	c.993T>C (p.Val331Val)	VUS	[27]

P—pathogenic, VUS—variant of unknown significance. CC—current case carrying the c.790A>G, p.Ile264Val missense variant.

## Data Availability

The data presented in this study are available on request from the corresponding author. The data are not publicly available due to privacy restrictions.

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
