# Peer review of "Rare Causes of Arterial Hypertension and Thoracic Aortic Aneurysms—A Case-Based Review"

_diagnostics, 2021, doi:10.3390/diagnostics11030446_

Round 1
Reviewer 1 Report
In this manuscript Encica et al. described a case of ascending aortic aneurysm (AA), which has been attributed to a variant of unknown significance of PRKG1 gene. Thus, the authors provided an overview on the current knowledge on genetic mutations and extraaortic features associated with aortic aneurysm.
This manuscript is well written, logically structured, and the included references are mostly appropriate.
Just few comments:
- Did the authors also investigate the involvement of other arterial districts? Was coronary angiography performed?
- clinical implication of the findings of this new genetic variant associated with AA should be better highlighted.
- A table summarizing the possible mechanism associated with arterial hypertension in patients with thoracic aortic aneurysm, which have been extensively described in section 3.2, would be advisable.
- I would suggest creating a sub-paragraph for therapy in this patients with PRKG1-associated AA.
Author Response
Answers to Reviewer 1:
This manuscript is well written, logically structured, and the included references are mostly appropriate.
We thank the Reviewer for the kind comment.
- Did the authors also investigate the involvement of other arterial districts? Was coronary angiography performed?
Thank you for the good observation, as arterial tortuosity including of the coronary arteries has been described in relation to other PRKG1 mutations. We have performed the angioCT, which did not show the involvement of other arteries. We did not perform coronary angiography before surgery, due the patient’s age and lack of coronary symptoms, as the genetic test results came afterwards. No obvious coronary abnormalities were noted during surgery, however coronary assessment will be included in the follow-up planning for the patient.
- Clinical implication of the findings of this new genetic variant associated with AA should be better highlighted.
We thank the Reviewer for the comment. We have included a new paragraph in the Conclusions section (page 14): Hypertension in a young patient with a TAA may point to a search of FBN-1, PRKG1 or other aorthopathy genes mutations. The PRKG1 variant in the patient described is of unknown significance and was not confirmed as pathogenic. Nevertheless, VUS in aortopathies genes can be low-penetrant “risk variants” conferring a predisposition which may be enhanced by other environmental, genetic or epigenetic risk factors. Hypertension and inflammation are important risk factors that should be searched for and addressed in order to minimize the risk (34),(35).
- A table summarizing the possible mechanism associated with arterial hypertension in patients with thoracic aortic aneurysm, which have been extensively described in section 3.2, would be advisable.
We thank the Reviewer for the suggestion. Please find enclosed the table with the mechanisms as suggested (page 11+12).
- I would suggest creating a sub-paragraph for therapy in this patients with PRKG1-associated AA.
We thank the Reviewer for the suggestion. We have created the sub-paragraph as suggested (page 6).

Reviewer 2 Report
Dear authors,
Your case presentation is interesting Please find attached some recommendations to clarify some aspects related to therapy.
Best regards

Author Response
Answers to Reviewer 2:
The case is interesting, debating the diagnosis and therapy of thoracic aortic aneurysms in young people and draws attention to the genetic factors involved in their etiology.
We thank the Reviewer for the kind comment.
Thus, there are some problems:
- On row 81 there is something wrong with dose of ivabradine – it should be administrated 5 mg twice daily and for perindopril indapamide the dose is 5/1.25 mg. Otherwise, it is curious why, in a patient with hypertension, ivabradine was preferred (probably for lowering heart rate) instead of a higher dose of betablocker that should have been useful for high blood pressure control.
We are grateful for the correction of the erroneously written medication dose. We have included the correct dose into the text (page 2).
Also thank you for the astute observation regarding the beta-blocker dose. Indeed, beta-blockers are preferred for blood pressure control in thoracic aortic aneurysms. Nevertheless, the patient had side effects to this medication class (erectile dysfunction), which made him reluctant to increase the dose for lowering the heart rate (that is why he was prescribed ivabradine).

Reviewer 3 Report
Interesting case report and review on thoracic aortic aneurysms.
I have only some minor suggestions to make in order to raise the competence of this article.
-Please clarify the echo parameters used to assess the aortic regurgitation as severe in your case.
-Are there differences regarding the localization (ascending, arch, descending aorta) and risk of acute thoracic syndrome of thoracic aortic aneurysms according to aetiology (genetic causes, hypertension, atherosclerosis, e.t.c.) ?
Author Response
Answers to Reviewer 3:
Interesting case report and review on thoracic aortic aneurysms.
We thank the Reviewer for the kind observation.
I have only some minor suggestions to make in order to raise the competence of this article.
-Please clarify the echo parameters used to assess the aortic regurgitation as severe in your case.
Thank you for helping us improve the observation. The echocardiographic exam revealed a tricuspid aortic valve with annular dilatation and moderate/severe aortic regurgitation (vena contracta 6 mm and PHT 360 msec). We have updated the text accordingly (page 2).
-Are there differences regarding the localization (ascending, arch, descending aorta) and risk of acute thoracic syndrome of thoracic aortic aneurysms according to aetiology (genetic causes, hypertension, atherosclerosis, e.t.c.) ?
Thank you for the kind suggestion. Indeed, there are differences regarding the localization and the risk of dissection and acute thoracic syndrome according to the etiology. We have included a new subchapter (3.3.- page 13) describing it.
